# The visual design of urban multimedia portals

Lin Wang[1], Yi Zhang [2]*

1 Department of Product Design, Sanming University, Sanming, China, 2 Division of Arts, Shenzhen University, Shenzhen, Guangdong, China

* zhangyiszu@163.com

## Abstract

In the visual design of a portal website, color is the first intuitive factor for users. It is relatively difficult for the designer of a city portal website to choose a color system that represents a city's unique color from among the many available options. Therefore, this study extracted a decision-making model of the urban color system, which can help decision-makers and designers choose among color systems, and then effectively design a portal website that conforms to local cultural attributes. The proposed method to solve the problem involved obtaining optimal color matching by performing weight analysis of colors through 123 sample color semantics, factor analysis, and a fuzzy analytic hierarchy process. Semantic analysis was used to classify colors into four categories of fashion, technology, calm, and dazzling. The fashion color matching scheme scored relatively high. Web page color matching schemes with a white background were popular, among which a white and green color matching scheme scored relatively high. At the same time, there are differences in color preferences between genders and cultures. This study is significant because it proposes a color decision model for portal websites, which provides a reference value that can also be applied to the selection of color schemes for other types of web pages in the future.

## 1. Introduction

On the Internet, a city portal is a window for delivering city images to online users irrespective of national geographic boundaries. Not only does it need to conform to local and native people's impressions of the city, it also needs to accurately convey the image of the city to users visiting the site for the first time. Therefore, the design of the city portal website should first highlight the city's unique culture, history, region, and other visual elements in the interface design, and at the same time, it should also be suitable for the user preferences of visitors from different countries.

In interface design, a practical interface layout can help users quickly obtain information. Color, as the visual identification element that accounts for the largest proportion of the interface area, and which is the first to catch the user's line of sight, will leave a deep impression on the user in the process of browsing the page. In addition, the Internet interface is different from a static screen, and the content of the Internet interface is updated from time to time. Therefore, we can only put constantly updated content and specifications in the corresponding position of the interface so that the information will not be cluttered. Thus, in the interface

**Data Availability Statement:** All experimental data files are available in the OSF database(https://osf. io/8zc7d/).

**Funding:** Sanming University. Research on Sustainable Factors of Social Innovation Design, Grant Number: KC21056S.The funders had no role

in study design, data collection and analysis, decision to publish, or preparation of the manuscript.

**Competing interests:** There are no conflcts of interest in the content of this article.

layout, color matching research about relatively stable colors and background colors occupying a large area in the interface can more accurately understand the color preferences of Internet users.

In the context of the rapid development of digital media, the color selection of city portals is not an easy task for designers [1]. In the design process, color selection needs to meet the psychological needs of the viewers and conform to local cultural characteristics. Guo [2] pointed out that color and visual design are the main visual languages to express the emotion and content of web pages. The layout of a web page is very important to the website's aesthetics and affects user preferences. Furthermore, web page color schemes are relatively subjective for design decisions, and there are cognitive differences among browsing customers. In addition, Lin [3] believes that the visual complexity of a website and the contrast of the background color of the graphics have a significant impact on consumers' emotional response, which will produce psychological changes when browsing shopping websites. On this basis, Won [4] pointed out that the tools of color are used in the development and design of web pages in a wide range of approaches, but only in large-scale research, and there is a lack of design approaches for special websites. Sandnes [5] proposed a visual detection framework that can tentatively adjust the color saturation but is separate from the human experience component. Park [6] thinks that in the city portal, website designers do not pay attention to the visitor's environment but to excellent design skills and propose that they should pay attention to the web design environment. Therefore, Nordhoff et al. [7] conducted research on the cultural background of visitors in 44 different countries, and conducted a design study. Kondratova and Goldfarb [8] conducted research on national color matching, believing that each country has its own specific color, and organized and analyzed the color of a sample group of 15 countries. On this issue, Rukshan [9] conducted research on cross-cultural web page color matching and believed that successful communication of different cultural groups requires different user interfaces, although research on cross-cultural website usability is necessary. Lukas [10] used a case study to segment composition in web pages and to design a study of color schemes used for the computational matching of logos. Relatively speaking, this research has limitations, and no research on a website's overall page design has been carried out. At the same time, Li [11] emphasized that aesthetic characteristics should always be consistent with the website function and believed that the layout and image design of a web page plays a very important role in the construction of the entire website. Chen [12] organized and analyzed it in a traditional theme website but did not develop a theoretical research framework. Yingxi's [13] research defined the main, secondary, and accent colors to carry out color experiments and studied the semantic language of color, which is also a new approach. On this basis, Hwang [14] initially proposed the concept of color intention through the study of color in web pages. Akhmadeeva et al. [1] proposed a method for automating the task of processing images and color schemes in web design for designers to use, limited to the fact that he considered color from the designer's point of view but did not consider the page visitor's perspective. Wu [15] found that music and color factors significantly affected participants' emotional responses and analyzed the sound and background music of web pages.

Color has social meanings in different cultures and backgrounds, and there are certain differences in feelings about color. Among them, Ok and Min [16] analyzed the color difference, color image, and hue perception of different groups of people, finding that there was no unified color or unified image among different cultural communities. In addition to cultural differences, Lilia [17] pointed out that the background color has a relationship with people's sense of trust, and color influences people's psychology. Ruse [18] argues that whether people are attracted to web pages is determined very quickly, pointing out that color combinations may be critical to successful website design. Cheng [19] found that color and price significantly

affected customer response while browsing commercial websites. Hsieh [20] analyzed the psychological effects on different consumers of color changes, especially in response to red and blue, when shopping, believing that color changes would positively and negatively affect consumers' purchasing psychology. Pelet and Papadopoulou [21] conducted research on the interaction of contrast, hue, and brightness in foreground and background colors to develop consumers' purchase intent. On this basis, Neda [22] conducted attitude assessments through online user questionnaires to collect demographic data affecting users' color preferences and interests and did not propose a systematic color scheme. Stenstrom [23] pointed out gender differences in human cognition related to navigation and stay when using web pages. Skulmowski et al. [24] believed that the saturation of color is important for the respondents' initial feeling and mainly conducted related experiments and research on changes in saturation. However, there is a lack of research on color categories. In the study of color design for different groups of people, Park [25] analyzed color-blind people to obtain relevant general design concepts and proposed an improvement plan for the color design of representative university mobile websites through the implementation of the group's research. Jiang [26] believed that an important factor of a well-designed web page is that the designer can create a good web page through the harmony and combination of colors, images, and characters. Kim [27] concluded that color has a certain influence on brand image. Kim [28] further researched the practice of color strategies in coffee shops.

Using a website is a human-computer interaction process, and its interactive environment [29] creates infinite possibilities for the design of websites, games, online platforms, and mobile applications. Lewandowska et al. believe that colors have similarities between different platforms. Their visual and functional properties affect the user experience. Oyibo and Vassileva [30] proposed that in travel website design for mobile devices, the grid layout of products and services provides a better hedonic user experience than the list layout. Among them, research has found that blue and green are more advantageous for the customer experience of travel websites. However, Tran Trung et al. [31] proposed that an orange interface is more user-satisfying than a gray interface. There are differences in user satisfaction between the two colors, indicating that people with differences in subdivided web page areas also differ in satisfaction with web page colors. Excessive textual information in the interaction process can elicit disgust. Baughan et al. [32] generally found relatively textual and minimalist designs in the process of browsing web pages, but people prefer highly complex websites with many colors, images, and texts. In the field of digital interaction, Hawlitschek et al. [33] found a relationship between user interface background color and user behavior during interaction. It is proposed that a red interface can enhance interaction behavior. Previous scholars have attempted cross-platform research on color. Okada and Castillo [34] combined Kansei Engineering's website comfort analysis and research, mainly on the layout of web pages, but lacked research and statistics on cultural samples. Kondratova and Goldfarb [35] pointed out culture-specific web interface design elements and conducted some research on the influence of culture on web design.

Previous research has found that people browse the web randomly. At the same time, as a gateway to a city, a portal website also has its own urban attributes, which include physical geography, humanities, and development vision. Previous scholars have rarely conducted research on the color design of portal websites, so they have discussed the random and uncertain habits of users and tried to build a color decision model for portal websites, to help designers and decision makers to find color matching that meets the psychological needs of browsers conveniently and appropriately when designing a portal website for a city. At the same time, this study proposes a decision-making framework for a color design for other different types of websites.

## 2. Theoretical background

### 2.1 Image color extraction

Screenshots of websites were taken and each webpage's color was extracted through a color hunter. By excluding adjacent pixels, a double-threshold method was used to determine RGB pixels and color names in a specific image, extracting significant informative colors for color, background color, and subject analysis [36]. The source data collected were selected by extracting digital color data using Photoshop 2020's Eyedropper tool. Similar colors of RGB color data were converted to select colors with relatively large hue differences, and induction and simplification were conducted [37].

### 2.2 Factor analysis

Factor analysis refers to statistical techniques for extracting common factors from groups of variables. Initial research obtain the relevant factors affecting academic performance through the analysis of students' academic performance. Subsequently, it was also widely applied to other different relationships. In conclusion, factor analysis can find hidden representative factors in many variables. Grouping variables of the same nature into one factor can reduce the number of variables, which can reduce the number of analyses and reduce the number of variables. In previous studies, factor analysis was quantitatively applied [38, 39]. In this study, through quantitative analysis of different color schemes, the weights between factors were reduced in dimensionality analysis to achieve classification and weighting.

### 2.3 Fuzzy analytic hierarchy process

The Analytic Hierarchy Process (AHP) was proposed by Professor T. L. Saaty of the University of Pittsburgh in the early 1970s [40]. AHP is a multi-objective decision analysis method that combines qualitative and quantitative analysis methods. The main idea of this method is to decompose a complex problem into several levels and several factors, make a comparative judgment on the importance of the two indicators, establish a judgment matrix, and calculate the maximum eigenvalue of the judgment matrix and the corresponding eigenvector. The weights of the importance of different schemes are obtained to provide a basis for the selection of the best scheme. However, in the calculation process of the AHP, the subjectivity of the survey experts is less considered in the method of constructing the judgment matrix. In order to solve the above shortcomings, a consistent fuzzy matrix was introduced into AHP. This method of combining the AHP with fuzzy mathematical theory is called the Fuzzy Analytic Hierarchy Process (FAHP), which aims to address the human subjective effect of ambiguity on the decision factors of a problem. Du proposed [41] to add fuzzy calculations to compare the value of FAHP and AHP because of the lack of conclusiveness of AHP. Lu [42] discussed the fuzzy judgment matrix based on FAHP, and proposed some important properties to verify the fuzzy consistency judgment matrix to analyze the weight ordering in FAHP.

**2.3.1 Mathematical theorem of fuzzy matrix.** Define an n-dimensional square matrix R:

$$R = \left(r_{ij}\right)_{n \times n} = \begin{bmatrix} r_{11} & r_{12} & \cdots & r_{1n} \\ r_{21} & r_{22} & \cdots & r_{2n} \\ \vdots & \vdots & \vdots & \vdots \\ r_{n1} & r_{n2} & \cdots & r_{nn} \end{bmatrix}$$

1. If the matrix R satisfies $0 \leq r_{ij} \leq 1, (i,j = 1,2,\cdots,n)$, then R is a fuzzy matrix;

2. The matrix R satisfies the condition 1, and $r_{ij}+r_{ji} = 1$, $(i,j = 1,2,\cdots,n)$, then R is a fuzzy complementary matrix;

3. If matrix R satisfies conditions 1 and 2, and: $r_{ii} = 0.5, (i = 1,2,\cdots, n)$; $r_{ij} = r_{ik}-r_{jk}+0.5, (i,j, k = 1,2,\cdots, n)$, then R is called a consistent fuzzy matrix.

### 2.3.2 Calculation steps of fuzzy AHP.

1. Fuzzy AHP score table
   Comparing each factor pairwise by the 0.1–0.9 scale method, it is stipulated that: The Table 1 is analytically defined in the definition of fuzzy AHP table [43].

2. Construction of Fuzzy AHP scoring matrix
   The above-mentioned 0.1–0.9 scale method is used to compare the evaluation factors in pairs, and the fuzzy AHP scoring matrix A is obtained:

$$A = \begin{bmatrix} a_{11} & a_{12} & \cdots & a_{1n} \\ a_{21} & a_{22} & \cdots & a_{2n} \\ \vdots & \vdots & \vdots & \vdots \\ a_{n1} & a_{n2} & \cdots & a_{nn} \end{bmatrix}$$

   Matrix A satisfies the requirements of the fuzzy complementary matrix, that is, $0 < a_{ij} < 1$; $a_{ij} + a_{ij} = 1$; $a_{ij} = 0.5$, $(i = j)$.

3. sum the matrix A row by row

$$a_i = \sum_{k=1}^{n} a_{ik}, (i, k = 1, 2 \cdots n)$$

4. Find the weight determinant WI of each factor

$$w_i = \frac{1}{n} - \frac{1}{2\alpha} + \frac{a_i}{n\alpha}$$

$$WI = [w_1\ w_2\ \cdots\ w_n]^T$$

**Table 1. Fuzzy AHP score table score.**

| Scoring criteria |
| --- |
| 0.1 Factor i is absolutely unimportant than factor j; |
| 0.2 Factor i is much less important than factor j; |
| 0.3 Factor i is less important than factor j; |
| 0.4 Factor i is slightly less important than factor j; |
| 0.5 Factor i is equally important than factor j; |
| 0.6 Factor i is slightly more important than factor j; |
| 0.7 Factor i is more important than factor j; |
| 0.8 Factor i is more important than factor j; |
| 0.9 Factor i is absolutely more important than factor j; |

Where $\alpha = \frac{n-1}{2}$.

5. Consistency CI test. Build the weight matrix W

$$w_{ij} = \alpha\left(w_i - w_j\right) + 0.5$$

$$W = \begin{bmatrix} w_{11} & w_{12} & \cdots & w_{1n} \\ w_{21} & w_{22} & \cdots & w_{2n} \\ \vdots & \vdots & \vdots & \vdots \\ w_{n1} & w_{n2} & \cdots & w_{nn} \end{bmatrix}$$

$$CI(A, W) = \frac{\sum_{i=1}^{n}\sum_{j=1}^{n}\left|w_{ij} - a_{ij}\right|}{n^2}$$

The smaller the CI value, the better the consistency. Generally, CI<0.1 means that the consistency requirements are met.

**2.3.3 Particle swarm optimization (PSO).** Due to the high subjectivity of expert scoring in the fuzzy analytic hierarchy process, we can use a particle swarm optimization (PSO) algorithm to correct the expert scoring matrix.

PSO, proposed by Dr. Eberhard and Dr. Kennedy in 1995, originated from research on the predation behavior of birds. Its core is to use individual information to conduct information analysis in the group so that the individuals in a group can obtain optimal decision-making. Through mutual learning among individuals, iterated step-by-step, the particles of the entire population will gradually tend to the optimal solution. Ivo et al. [44] performed optimization on PSO and its inapplicability in constrained optimization problems using dynamic objective constraint handling methods.

PSO is initialized as a group of random particles (random solution). Then the optimal solution is found iteratively. The particle updates itself by tracking two "extremes" in each iteration. After finding these two optimal values, the particle updates its velocity and position by the following formula.

$$V_{i+1} = V_i + c_1 \times \text{rand}(0 \sim 1) \times (\text{pbest}_i - x_i) + c_2 \times \text{rand}(0 \sim 1) \times \left(\text{gbest}_i - x_i\right)$$

$$x_{i+1} = x_i + V_i$$

i = 1, 2,. . ., M, M is the total number of particles in the group; $V_i$ is the velocity of the particle; pbest is the individual optimal value; gbest is the optimal global value; rand (0~1) is between (0, 1) between random numbers; $X_i$ is the current position of the particle. $c_1$ and $c_2$ are learning factors, usually $c_1 = c_2 = 2$. In each dimension, the particle has a maximum speed limit $V_{max}$; if the speed of a dimension exceeds the set $V_{max}$, then the speed of this dimension is limited to $V_{max}$.

## 2.4. Summary of special vocabulary

The abbreviations of some special vocabulary involved in Table 2 are organized and summarized here to facilitate understanding the full text.

**Table 2. Fuzzy AHP score.**

| Shorthand | Full name |
|---|---|
| FAHP | Fuzzy Analytic Hierarchy Process |
| PSO | Particle Swarm Optimization |
| KMO | Kaiser Meyer Olkin |
| SPSS | Statistical Product Service Solutions |
| CR | Consistency Ratio |
| CI | Consistency Index |

## 3. Research content and results

This research project combined portal user sentiment analysis with image color extraction technology, which can be divided into three stages. In the first stage, the foreground and background colors of 123 portal websites were obtained through color extraction, as shown in Fig 1.

After data sorting and website extraction, the collected color data were sorted and classified. Ten colors were summarized (Table 3) (white, gray, black, red, orange, yellow, green, blue, purple, and brown). A total of 100 matching combinations of 10 main colors were obtained in pairs, and a total of 55 web page colors in Fig 2 were obtained by removing the repeated pairs. Adobe Photoshop was used for color extraction and web demo design in this study.This research project, participants were randomly selected, the experimental procedures were reported and their verbal consent was obtained. A total of 80 people participated in the experiment, including 38 males and 42 females. The respondents ranged in age from 18 to 58 and included city dwellers, students, and university faculty.

In the second stage, the Litt scale was used to score the 55 factors, and then SPSS was used for factor analysis to analyze the 55 color matching schemes. The result was that the factors were clustered into four components, as shown in Table 3. At the same time, factor analysis was used to obtain the weight value between color matching to obtain the best six color schemes (White/White, White/Gray, White/Red, White/Yellow, White/Green, White/Blue). Before using factor analysis, the correlations between the data need to be calculated using the Kaiser Meyer Olkin (KMO) significance coefficient and Bartlett's sphericity test: results of

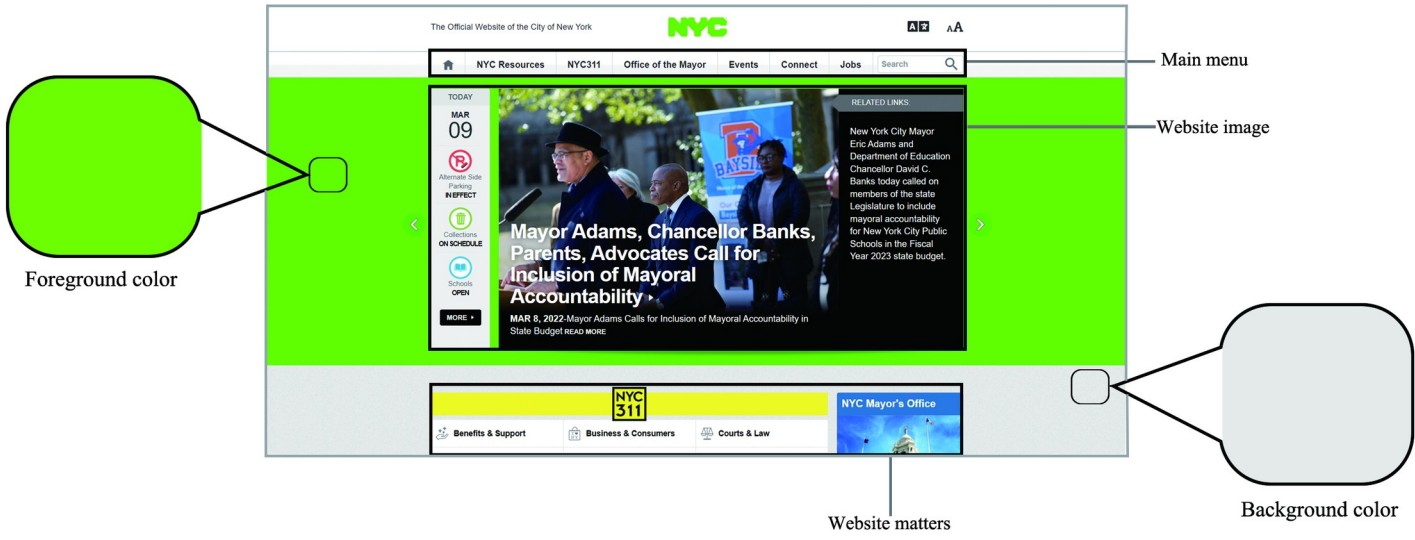

**Fig 1. Color extraction of multimedia portals.**

**Table 3. Main 10 colors RGB.**

| Color name | RGB | | |
|---|---|---|---|
| White | 255 | 255 | 255 |
| Gray | 233 | 232 | 231 |
| Black | 0 | 0 | 0 |
| Red | 231 | 31 | 25 |
| Orange | 253 | 119 | 77 |
| Yellow | 239 | 234 | 58 |
| Green | 117 | 187 | 42 |
| Blue | 0 | 39 | 111 |
| Purple | 109 | 52 | 101 |
| Brown | 129 | 81 | 28 |

KMO > 0.7 [45] and p< 0.01 would confirm that the indicators are correlated. In the results, the KMO value is 0.716, and the p value of the Bartlett sphericity test is 0.000, indicating that the sample and the factors are correlated, indicating that these data are suitable for factor analysis.

The results (Table 4) showed that white backgrounds were relatively popular. The table component analysis (Table 5) color pairing clustering found four categories: fashion, technology, dazzling, and calm. The fashion score was generally high, and the highest six color schemes (White/White, White/Gray, White/Red, White/Yellow, White/Green, White/Blue) were obtained.

In the third stage, FAHP was used to analyze the six optimal color schemes of web pages in the general web page structure, and Adobe Photoshop was used to create a web page color matching model, as shown in Fig 3. Then, the color scheme with six colors was tested on the computer according to the color mode of item numbers 1 to 6. This was scored indoors by changing sheets every 30 seconds. Each test subject scored on the FAHP scoring table when they saw the first reaction to the picture. The weights were obtained through the calculation.

In using FAHP to calculate the weights, fuzzy calculation and PSO were introduced because of the limitations of human subjective factors. PSO addresses the problem of human subjective

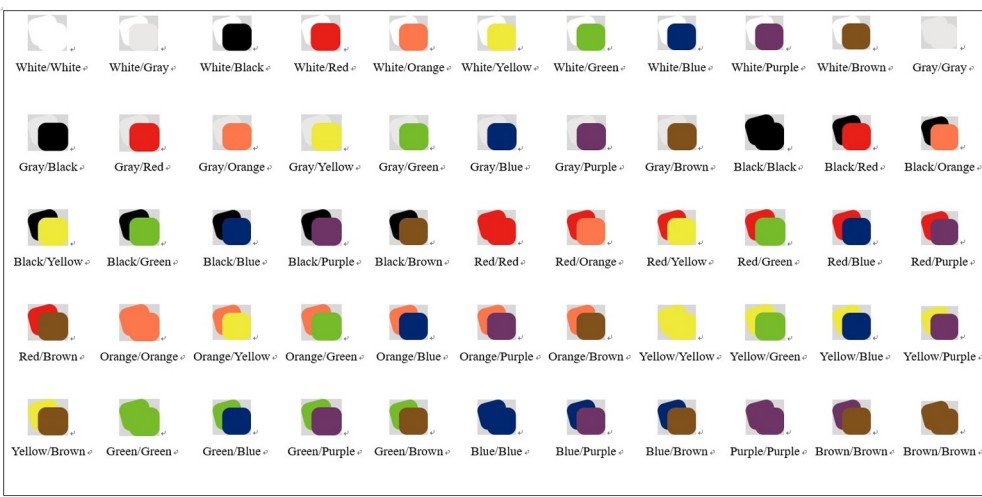

**Fig 2. Two-color color scheme diagram.**

**Table 4. One-factor analysis score table.**

| Color | average value | standard deviation | number of copies |
|---|---|---|---|
| 1.White/White | 4.18 | 1.682 | 80 |
| 2.White/Gray | 4.19 | 1.584 | 80 |
| 3.White/Black | 3 | 1.669 | 80 |
| 4.White/Red | 4.59 | 1.573 | 80 |
| 5.White/Orange | 3.79 | 1.481 | 80 |
| 6.White/Yellow | 4.06 | 1.529 | 80 |
| 7.White/Green | 4.63 | 1.554 | 80 |
| 8.White/Blue | 4.63 | 1.767 | 80 |
| 9.White/Purple | 3.9 | 1.62 | 80 |
| 10.White/Brown | 4 | 1.714 | 80 |
| 11.Gray/Gray | 3.43 | 1.589 | 80 |
| 12.Blue/Purple | 3.1 | 1.79 | 80 |
| ⋮ | ⋮ | ⋮ | ⋮ |
| 55.Brown/Brown | 2.59 | 1.719 | 80 |

bias. The main judgment criterion is to check whether the judgment matrix has the consistency standard CR < 0.1 as the basis, so it is necessary to ensure that consistency can be passed in the calculation process.

The results show that the multimedia city portals can be clustered into four categories (as shown in Table 4), and the four categories can be divided into fashion, technology, calm, and

**Table 5. Single-factor analysis component table.**

| | 1 | 2 | 3 | 4 |
|---|---|---|---|---|
| 1.Yellow/Brown | 0.671 | 0.092 | -0.144 | -0.263 |
| 2.Yellow/Blue | 0.663 | -0.04 | -0.093 | -0.429 |
| 3.Orange/Blue | 0.649 | -0.01 | -0.257 | -0.126 |
| 4.Gray/Yellow | 0.647 | -0.076 | 0.211 | -0.075 |
| 5.Gray/Orange | 0.64 | 0.182 | -0.043 | -0.143 |
| 6.Black/Yellow | 0.631 | -0.02 | 0.299 | -0.174 |
| 7.Yellow/Green | 0.62 | 0.036 | -0.213 | -0.121 |
| 8.Red/Purple | 0.614 | 0.316 | -0.167 | -0.149 |
| 9.Orange/Orange | 0.6 | 0.147 | -0.114 | 0.282 |
| 10.Green/Purple | 0.591 | -0.339 | -0.208 | 0.07 |
| 11.Green/Blue | 0.587 | -0.289 | -0.168 | 0.085 |
| 12.Gray/Brown | 0.584 | 0.031 | 0.227 | -0.023 |
| 13.Black/Orange | 0.106 | 0.246 | 0.526 | -0.306 |
| 14.Black/Red | 0.197 | 0.318 | 0.52 | -0.041 |
| 15.Gray/Gray | 0.39 | -0.053 | 0.512 | 0.101 |
| 16.Black/Brown | 0.359 | 0.062 | 0.483 | -0.175 |
| 17.Black/Black | 0.339 | 0.065 | 0.446 | 0.146 |
| 18.White/Gray | -0.113 | 0.163 | 0.415 | 0.293 |
| 19.White/Black | 0.094 | 0.071 | -0.345 | 0.051 |
| 20.White/Brown | 0.203 | -0.265 | 0.291 | 0.168 |
| 21.White/Orange | 0.177 | 0.16 | -0.279 | 0.211 |
| ⋮ | ⋮ | ⋮ | ⋮ | ⋮ |
| 55.Brown/Brown | 0.301 | -0.219 | 0.091 | 0.362 |

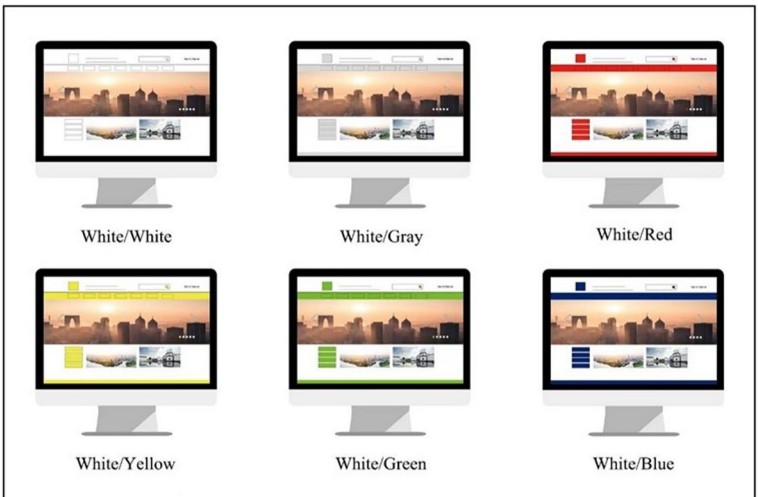

**Fig 3. Color map of the multimedia portal website.**

dazzling through the color semantic method [46]. The fashion category accounted for 30 categories; the technology category accounted for nine; the calm category accounted for ten; the dazzling category accounted for five. The representative fashion categories are Yellow/Brown and Yellow/Blue and most of the white backgrounds are basically in this category. Among them, the technology category includes Blue/Red, and the calm category includes factors such as Black/Blue, Black/Orange, Black/Red, Gray/Gray, and Black/Brown. Tables 3 and 4 show that color matching of web pages with a white background is popular, and the scores of technology categories were relatively high. It was concluded that the White/Green color scheme scored relatively high in this category.

In the Table 6, The weight matrix was obtained in the order of White/Green > White/White > White/Blue > White/Gray > White/Red > White/Yellow by performing FAHP calculation on the extraction of the optimal color of the six color schemes.

## 4. Discussion and evaluation

In this study, optimal color matching was obtained by color extraction and factor analysis, and then the combination of software and common portal websites was used to obtain the optimal portal website combination. The six optimal color schemes were analyzed through FAHP to obtain the optimal scheme weight value. A previous study by Guo [2] pointed out that users' least favorite color combinations were red/yellow and red/lime green, which were unpopular

**Table 6. FAHP analysis table.**

|  | White/White | White/Gray | White/Red | White/Yellow | White/Green | White/Blue | Weights(wi) |
|---|---|---|---|---|---|---|---|
| White/White | 0.5 | 0.5 | 0.5 | 0.7 | 0.3 | 0.35 | 0.1567 |
| White/Gray | 0.5 | 0.5 | 0.5 | 0.7 | 0.3 | 0.45 | 0.1633 |
| White/Red | 0.5 | 0.5 | 0.5 | 0.6 | 0.3 | 0.45 | 0.1567 |
| White/Yellow | 0.3 | 0.3 | 0.4 | 0.5 | 0.25 | 0.3 | 0.1033 |
| White/Green | 0.7 | 0.7 | 0.7 | 0.75 | 0.5 | 0.6 | 0.23 |
| White/Blue | 0.65 | 0.55 | 0.55 | 0.7 | 0.4 | 0.5 | 0.19 |

Weight matrix for calculation after correction: Portal Color Matching: Consistency CI = 0.025

among all participants in the experiment, and the conclusion was basically the same. Among the most comfortable colors, cyan/black and cyan/gray were considered as comfortable colors. This study considered six color schemes on white (White/White, White/Gray, White/red, White/Yellow, White/Green, White/Blue) that most people liked, although there were differences. Compared with previous studies, this study provides a more refined analysis of the impact of web page structure on people's perception of the website. The conclusion is that white is generally more popular as the background color of the portal website. However, there may be some differences in the perception of color related to regional, cultural, or gender factors, so the FAHP method was used to compare the relevant factors of the participants.

## 4.1 Differences in color perception by gender

Some scholars have studied color-related gender differences in the interior design process through color lexicalization [47]. Zheng et al. [48] believed that the color design of symbols is very important, and there are gender differences in their feelings, which is consistent with relevant conclusions in psychology that gender affects color selection tendency. However, Gou et al. [2] did not consider gender in the research process and believed gender has little effect on color perception. Hence, there has been discussion on whether there are gender differences in the emotional impact of color. First, the survey was divided into two groups (6 women and 5 men), and performed FAHP weight analysis on the previously determined six optimal color matching websites (Fig 3).

The weighted analysis for different genders is shown in Table 7. There are certain similarities and differences in the optimal color matching schemes of multimedia portal websites. Both men and women like White/Green the most. There was a difference in that women liked White/Red with a score of 0.1867, while the men's White/White score was higher at 0.1933. Both men and women scored the White/Yellow colorway lowest. The color identity of White/Blue and White/Gray was relatively the same, and the score was relatively high.

## 4.2 Comparison of color pairs of different cultural backgrounds

In discussing whether different cultural backgrounds affect the color perception of web pages, there has been related research in three different countries [49]. Comparative analysis has been carried out, although it had certain limitations because of the use of relatively few colors. Some scholars believe that in the process of web design, it is very challenging for designers to design a common international color matching website that would be liked by many peoples and cultures around the world [50]. Therefore, this study conducted weight analysis and discussion through six previously determined color schemes of foreground and background colors. The survey conducted FAHP questionnaires for international students. Select students from Japan, the United Kingdom, and China completed the survey to study the different influences of various cultural backgrounds on perceptions of website colors.

**Table 7. Gender FAHP middle layer weight table.**

|  | Women | Men |
|---|---|---|
| White/White | 0.12 | 0.1933 |
| White/Gray | 0.1533 | 0.1733 |
| White/Red | 0.1867 | 0.1267 |
| White/Yellow | 0.0933 | 0.1133 |
| White/Green | 0.2533 | 0.2067 |
| White/Blue | 0.1933 | 0.1867 |

Table 8 shows certain differences in the color perception of websites between different cultures. Japanese people gave a low score of 0.06 for White/Red backgrounds and do not like red very much. In the United Kingdom and China, people gave higher scores for red at 0.2333 and 0.1567, respectively. The three countries gave relatively high weights for White/Green color matching. This shows that people in these three countries feel better about White/Green. In the comparison of the three countries, the weights of the two color schemes, White/Green and White/Blue, were generally higher than 0.16 to 0.23. This shows that different cultures also have color matching schemes that the public likes.

This study still has certain limitations. The sample size for the selection of color schemes was still relatively small, and larger samples are needed to more accurately represent the national culture. In addition to gender and culture, age may also influence color perception. Therefore, this study selected six optimal color matching schemes of participants between the ages of 18 and 58 years old in the previous factor analysis. The sample size range selected in the cultural and gender comparison process is still too small, and it is hoped that it will be expanded in future research.

## 4.3 Evaluation

For the evaluation of different color matching methods, comparison analysis of this color selection model and other models was used. Previous research used Colormind, a website that uses neural networks for color assignment. Various other methods have been used in research in the field of automatic color generation [51]. You Weitao [52] used a clustering method to select advertising colors and obtained color preferences by collecting a certain number of advertisement pictures. A limitation of this method is that only three types of advertisements can be determined when the selection of advertisements is too broad. Gu and Lou [53] proposed an automatic color-making model for web pages, analyzed the collected web page samples, and used a regression model to optimize the web page color blocks to obtain the optimal color. A limitation of this method is that because there are many types of websites, the scope is too large. To verify the effectiveness of the method, three different methods were tested. They are our own method, a data-driven method [53], and Colormind.

Additionally, we ran FAHP tests to analyze the effectiveness of different methods for producing final web pages. The same web page structure was prepared to extract the colors obtained by different scoring methods. Five graphic design teachers and six laypeople were asked to rate the match between images and keywords using a 5-point Likert scale. As shown in Fig 4 the analysis mainly tested and scored the three methods for professionals and non-professionals. The scores obtained by the colors produced by our method are relatively high, and the same results can be seen from both the professional and non-professional groups.

**Table 8. FAHP weight table of different cultures.**

|  | Japan | UK | China |
|---|---|---|---|
| White/White | 0.22 | 0.16 | 0.1567 |
| White/Gray | 0.18 | 0.1267 | 0.1633 |
| White/Red | 0.06 | 0.2333 | 0.1567 |
| White/Yellow | 0.1533 | 0.12 | 0.1033 |
| White/Green | 0.2067 | 0.1933 | 0.23 |
| White/Blue | 0.18 | 0.1667 | 0.19 |

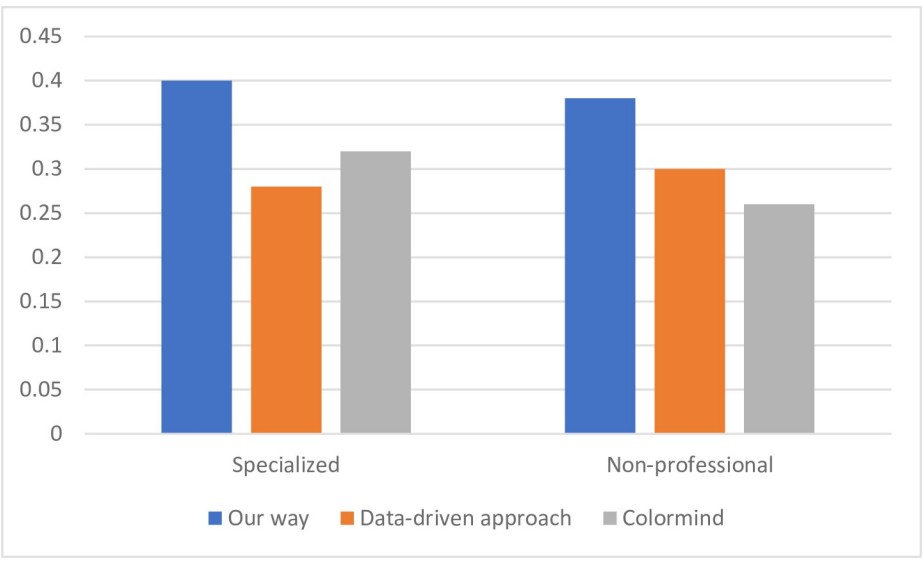

**Fig 4. Method comparison analysis chart.**

## 5. Conclusions

This study used a survey to collect the foreground and background colors of 123 portal websites, and used 55 kinds of portal website color matching schemes as the experimental model. Factor analysis was used to perform clustering and weight analysis on different color matching schemes, and the participants' responses to different colors were obtained to determine the optimal portal color scheme. On the basis of the optimal structure of the portal website, FAHP was used to analyze the webpages of six different color matching schemes and a single structure was obtained to perform pairwise comparisons, according to the obtained weighted values. According to the results, the following conclusions were reached. There are 55 kinds of color matching schemes that can be grouped into four categories of color matching, using semantic analysis [46]. The four types of adjectives are fashion, technology, calm, and dazzling. The factor analysis results showed that users of portal websites prefer fashionable color matching websites. Six websites with white backgrounds have higher scores, specifically, White/White, White/Gray, White/red, White/Yellow, White/Green, and White/Blue. Studies have found that Yellow/Brown and Yellow/Blue are highly related to fashion, and Blue/Brown and Red/Yellow are related to technology. Styles are highly correlated, with Black/Blue and Black/Orange being considered relatively calm, and Purple/Purple and Green/Green being considered dazzling colors. It is worth noting that in the research, low scores in the Brown/Brown and Purple/Brown color experiments indicated that efforts were made to avoid these colors in the portal design process. The six colorways with the highest scores obtained through the initial factor analysis were designed in the web page. Using FAHP showed that White/Green > White/White > White/Blue > White/Gray > White/Red > White/Yellow in terms of weight ordering. This research provides a certain reference value for future designers in decision-making about the layout and color matching of urban portal websites. At the same time, the study found that gender influences color preferences. Women prefer red, website color matching prefers White/Red, and men score higher in the White/White ratio. Research has found that White/Green are preferred color schemes for both men and women. In the comparison of different cultures, White/Red was less popular with Japanese people, while the opposite was found in the UK. White/Blue and White/Green are popular in three countries with different cultural backgrounds.

This study proposes a decision-making model of an urban color system, which helps decision-makers and designers to choose among color systems and effectively design a multimedia portal website in line with local cultural attributes. Differences in color matching directly affect people's emotional responses. This research makes an innovative contribution to website color matching selection schemes. Research on users' psychological responses to products needs to consider emotional factors [54–57]. Some scholars have used big data to classify color matching schemes [53–58]. Research has also been conducted using cloud computing, neural networks, and other method applications in hotel service social media to improve the user experience [59–61]. Among them, other application methods have reference value for color emotion research, which can be applied in future research.

The research shows that white and green had the highest scores in the weight analysis, which indicates that users prefer these the most when browsing city portals. However, this study has certain limitations. In the choice of color matching, because there are many kinds of colors, the middle and similar colors are selected in the process of selecting colors. In addition, because the regional difference survey was limited by the limited sample of the survey population, it was impossible to extract the preferences of a finer population, and it is hoped that this can be improved in future research. In this study, a new method is proposed for the formation of optimal color schemes in multimedia portals, and the research results can be used as a reference for designers. Although this research examined portal multimedia web pages, the results can be used as a theoretical and practical basis for the development and design of other types of web pages, and have certain reference value and significance for developers and designers in related research fields.

## Acknowledgments

Thanks to Ruan Chenglu, Zhu Minghui and Sanming Integrated Medicine Hospital for their help on this project.

## Ethics statement

This study was approved by the Ethics Committee of Sanming Integrated Medicine Hospital (approval number: 2022-KY-05), approved the conduct of the questionnaire, and all subjects gave informed consent, obtained oral consent and signed by the participants under the supervision of the ethics committee.

## Author Contributions

**Conceptualization:** Lin Wang.

**Formal analysis:** Lin Wang, Yi Zhang.

**Investigation:** Lin Wang, Yi Zhang.

**Methodology:** Lin Wang.

**Project administration:** Lin Wang, Yi Zhang.

**Supervision:** Yi Zhang.

**Writing – original draft:** Lin Wang, Yi Zhang.

**Writing – review & editing:** Lin Wang, Yi Zhang.

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
