## [Decision Letter · Decision Letter 0]

2 Aug 2022

PONE-D-22-14953Research on Color Behavior of Urban Multimedia PortalPLOS ONE

Dear Dr. Zhang,

Thank you for submitting your manuscript to PLOS ONE. After careful consideration, we feel that it has merit but does not fully meet PLOS ONE’s publication criteria as it currently stands. Therefore, we invite you to submit a revised version of the manuscript that addresses the points raised during the review process.

We look forward to receiving your revised manuscript.

Kind regards,

Brij Bhooshan Gupta

Academic Editor

PLOS ONE

Journal Requirements:

2. In line with the principles expressed in the Declaration of Helsinki, we expect all research involving human participants and/or medical data to have been approved by the authors' Institutional Review Board (IRB) or by equivalent ethics committee(s). If the need for ethical approval is waived, this should be formally confirmed by a suitable committee generally before the start of the study. Please clarify whether there was any ethical oversight of the study." 3) "Please clarify in the Methods section how the participants were recruited, and how they provided consent." 4) "Please include information in the Methods section of whether real website portals were used, and if so, provide links to these.

4. Thank you for including your ethics statement:  "The questionnaires in this study were conducted anonymously with the oral consent of the respondents and therefore met the requirements of the Chinese mainland for ethical investigations".  

a. For studies reporting research involving human participants, PLOS ONE requires authors to confirm that this specific study was reviewed and approved by an institutional review board (ethics committee) before the study began. Please provide the specific name of the ethics committee/IRB that approved your study, or explain why you did not seek approval in this case.

“Funding Sanming University. Research on Sustainable Factors of Social Innovation Design, Grant Number: KC21056S.”

6. Thank you for stating the following in your Competing Interests section: 

“There are no conficts of interest in the content of this article.”

7. In your Data Availability statement, you have not specified where the minimal data set underlying the results described in your manuscript can be found. PLOS defines a study's minimal data set as the underlying data used to reach the conclusions drawn in the manuscript and any additional data required to replicate the reported study findings in their entirety. All PLOS journals require that the minimal data set be made fully available. For more information about our data policy, please see http://journals.plos.org/plosone/s/data-availability.

8. Please ensure that you refer to Figure 2 in your text as, if accepted, production will need this reference to link the reader to the figure.

9. We note you have included a table to which you do not refer in the text of your manuscript. Please ensure that you refer to Table 1, 2, 5, 6 and 7 in your text; if accepted, production will need this reference to link the reader to the Table.

Reviewers' comments:

Reviewer's Responses to Questions

**Comments to the Author**

1. Is the manuscript technically sound, and do the data support the conclusions?

Reviewer #1: Partly

Reviewer #2: Yes

2. Has the statistical analysis been performed appropriately and rigorously? 

Reviewer #1: No

Reviewer #2: Yes

3. Have the authors made all data underlying the findings in their manuscript fully available?

Reviewer #1: Yes

Reviewer #2: Yes

4. Is the manuscript presented in an intelligible fashion and written in standard English?

Reviewer #1: No

Reviewer #2: Yes

5. Review Comments to the Author

Reviewer #1: The technical contribution of this research is not adequately described in the abstract. Kindly revise it along with the following comments:

-Improve the accuracy of the data and provide adequate justification.

-The findings part is lacking, thus I advise the writers to expand it and contrast it with the current methods.

-This paper is beautifully written, although the present draught still contains a few errors. I recommend that the authors thoroughly proofread this essay and fix any mistakes in the revised version.

-To raise the caliber of literature, some current work that is closely relevant may be mentioned. For instance, it is expected that the authors will include the execution duration of the suggested algorithm in the change. i Siggest a few to improve the quality of the literature-Using an Ontology-Based Neural Network and DEA to Discover Deficiencies of Hotel Services, Scholar Recommendation Based on High-Order Propagation of Knowledge Graphs,Modelling of the Cloud Service Quality Factors Using ISM, Security and Privacy of Cloud-Based Online Online Social Media: A Survey

Additionally, references must include all pertinent data, including page numbers, journal or conference names, volume numbers, issues, etc.

Reviewer #2: • The abstract needs to be rewritten to point out significance and impact of the paper.

• In the related work, it is recommended to refer the contribution made by the researchers and the novelty of the research. However, the author does not mention that.

• I recommend that the authors add some more current articles to improve the paper's overall quality. The preparation of a comparative analysis of the current publications on this subject should also be included.

• Avoid presenting with lengthy paragraph.

• Paper needs to polish and provide a detailed explication of theoretical/systematic aspects behind this paper.

• Add more discussion on Color Behavior for that include quality research paper in the revised version.

• Some more clarification regarding the motivation and challenges of the research.

• Notations and acronyms used in this paper should be summarized in a table to organize this paper in a better way.

• Improve the quality of figures and explain those properly.

• Finally, a final proof-reading is highly suggested, in order to correct some typos.

6. PLOS authors have the option to publish the peer review history of their article (what does this mean?). If published, this will include your full peer review and any attached files.

Reviewer #1: No

Reviewer #2: No

---

## [Author Response · Author response to Decision Letter 0]

16 Nov 2022

Reviewer #1: 

-The technical contribution of this research is not adequately described in the abstract.

In response to your suggestion that the abstract part is not complete, I have rewritten the abstract. It is now structured to present the study purpose and question, method of use, results, and conclusion. Below is part of my newly written summary. 

“In the visual design of the portal website, color is the first intuitive factor for users. It is relatively difficult for the designer of a city portal website to choose a color system that represents a city's unique color from among the many options. Therefore, this study extracted a decision-making model of the urban color system, which can help decision-makers and designers to choose among color systems, and then effectively design a portal website that conforms to local cultural attributes. The proposed method to solve the problem involved obtaining optimal color matching by performing weight analysis of colors through 123 sample color semantics, factor analysis, and a fuzzy analytic hierarchy process. Semantic analysis was used to classify colors into four categories of fashion, technology, calm, and dazzling. The fashion color matching scheme scored relatively high. Web page color matching schemes with a white background were popular, among which a white and green color matching scheme scored relatively high. At the same time, there are differences in color preferences between genders and cultures. This study is significant because it proposes a color decision model for portal websites, which provides a reference value that can also be applied to the selection of color schemes for other types of web pages in the future.

-Improve the accuracy of the data and provide adequate justification.

Thank you very much for pointing out the problem of data accuracy. I have re-calculated all the data obtained from the survey. In terms of the accuracy of factor analysis, it was indeed found that the original data were biased, so the Kaiser–Mayer–Olkin (KMO) value was adjusted from 0.710 to 0.616. A KMO value of 0.616 can still confirm the applicability of the data. The construct validity was measured using Bartlett’s test of Sphericity and the KMO measure of the sampling adequacy of individual variables. KMO overall should be 0.6 or over to perform factor analysis [1]. In reliability studies, alpha is used when assessing internal consistency, which indicates that scale items are consistent with each other and contribute in the same direction to the scale. In order for the scale to be reliable, the alpha value must be higher than 0.70 [2,3]. In this study, the alpha value was 0.911. Therefore, this result is reliable. Thank you again for your question about the accuracy of the article's data. Later, we will upload all the data of the article to a database for sharing, which will be convenient for scholars to conduct research in the future.

1. Rajapathirana RPJ, Hui Y. Relationship between innovation capability, innovation type, and firm performance. J Innov Knowl. 2018;3: 44-55. doi: 10.1016/j.jik.2017.06.002, PubMed: WOS. 000425894600005.

2. Polit FD, Beck TC. Essentials of nursing research appraising evidence for nursing practice. 7th ed. Philadelphia: Wolters Kluwer Health - Lippincott Williams & Wilkins; 2010.

3. Yeşim P. Akyol Brief emergency department patient satisfaction scale (BEPSS): Turkish validity and reliability study 2021.

-The findings part is lacking, thus I advise the writers to expand it and contrast it with the current methods. 

Thank you for pointing out that you found some problems. I found relevant literature to expand the conclusion section. For the comparative analysis and evaluation of different methods and our model method, we wrote the following in the evaluation Section 4.3: 

“For the evaluation between different color matching methods, comparative analysis of this color selection model with other models was used. Previous research used Colormind, a website that uses neural networks for color assignment. Various other methods have been in research in the field of automatic color generation [1]. You Weitao [2] used a clustering method to select advertising colors, and obtained color preferences by collecting a certain number of advertising pictures. A limitation of this method is that only three types of advertisements can be determined when the selection of advertisements is too broad . Gu and Lou proposed a web page automatic coloring model, analyzed the collected web page samples, and used a regression model to optimize the web page color blocks to obtain the optimal color[3]. A limitation of this method is that because there are many types of websites, the scope is too large. To verify the effectiveness of the method, three different methods were tested. They are our own method, a data-driven method [3], and Colormind.”

“Additionally, we ran FAHP tests to analyze the effectiveness of different methods for producing final web pages. The same web page structure is prepared to extract the colors obtained by different scoring methods. Five graphic design teachers and six laypeople were asked to rate the match between images and keywords using a 5-point Likert scale. The analysis mainly tested and scored the three methods for professionals and non-professionals. The scores obtained by the colors produced by our method are relatively high, and the same results can be seen in both the professional and the non-professional groups.”

1. Zhang R, Zhu JY, Isola P, Geng XY, Lin AS, Yu TH, et al. Real-time user-guided image colorization with learned deep priors. ACM Trans Graph. 2017;36: 1–11. doi: 10.1145/3072959.3073703. 

2. You WT, Sun LY, Yang ZY, Yang CY. Automatic advertising image color design incorporating a visual color analyzer. Comput Languages. 2019;55. doi: 10.1016/j.cola.2019.100910. 

3. Gu Z, Lou J. Data driven webpage color design. Comput Aid Des. 2016;77: 46-59. doi: 10.1016/j.cad.2016.03.001.

-This paper is beautifully written, although the present draught still contains a few errors. I recommend that the authors thoroughly proofread this essay and fix any mistakes in the revised version.

Thank you very much for your affirmation. I will perform a new proofreading and polishing of the article to resolve the errors in the article that you pointed out. We hired a professional editing agency to revise the errors in the article and sentences to increase the readability of the article.

-To raise the caliber of literature, some current work that is closely relevant may be mentioned. For instance, it is expected that the authors will include the execution duration of the suggested algorithm in the change. i Siggest a few to improve the quality of the literature-Using an Ontology-Based Neural Network and DEA to DISCover Deficiencies of Hotel Services, Scholar Recommendation Based on High-Order Propagation of Knowledge Graphs,Modelling of the Cloud Service Quality Factors Using ISM, Security, and Privacy of Cloud-Based Online online Social Media: A Survey

Thank you very much for the several articles you recommended, I read the articles carefully and they were very helpful in improving the quality of the manuscript. Hence, we cited them in the introduction. “

This study proposes a decision-making model of an urban color system, which can help decision-makers and designers to choose among color systems, to effectively design a multimedia portal website in line with local cultural attributes. Differences in color matching directly affect people's emotional responses. This research makes an innovative contribution to website color matching selection schemes. Research on users’ psychological responses to products needs to consider emotional factors [2–5]. Some scholars have used big data to classify color matching schemes [1,6]. Research has also been conducted using cloud computing, neural networks, and other method applications in hotel service social media to improve the user experience [7-9]. Among them, other application methods have reference value for color emotion research, which can be applied in future research.

1. Gu Z, Lou J. Data driven webpage color design. Comput Aid Des. 2016;77: 46-59. doi: 10.1016/j.cad.2016.03.001.

2. Zhao Y. Emotional design of children’s medical devices based on color perception. J Mach Des. 2019;36: 142-144. 

3. Won S, Westland S. Requirements capture for colour information for design professionals. Color Res Appl. 2018;43: 387-395. doi: 10.1002/col.22198.

4. Wang Y, Cui M, Li G. Study on product color design based on human factors. Packaging Eng. 2013;34: 53-56. 

5. Wang K, Yu S, Yue W, Lu C. Color design system for computer-aided industrial design. J Compute-Aid Des Graph. 2004;16: 1425-1429. 

6. Li P, Li T, Wang X, Zhang S, Jiang Y, Tang Y. Scholar recommendation based on high-order propagation of knowledge graphs. Int J Semant Web Inf Syst. 2022;18: 1-19. doi: 10.4018/IJSWIS.297146.

7. Yadav US, Gupta BB, Peraković D, Peñalvo FJG, Cvitić I. Security and privacy of cloud-based online online social media: A survey. Sustainable management of manufacturing systems in Industry 40. Ea j/Springer Innovations in Communication and Computing. 2022: 213-236.

8. Chiang T-A, Che ZH, Huang Y-L, Tsai C-Y. Using an ontology-based neural network and DEA to DISCover deficiencies of hotel services. Int J Semant Web Inf Syst. 2022;18: 1-19. doi: 10.4018/IJSWIS.306748.

9. Agarwal R, Dhingra S. Modelling of the cloud service quality factors using ISM. Int J Cloud Appl Comput. 2022;12: 1-16. doi: 10.4018/IJCAC.295241.

-Additionally, references must include all pertinent data, including page numbers, journal or conference names, volume numbers, issues, etc.

Thank you very much for the problem you pointed out. We will proofread it again to ensure that the references contain all the necessary information.

Reviewer #2:

• The abstract needs to be rewritten to point out significance and impact of the paper.

I have rewritten the abstract, using the structure of stating the purpose and question, the method uses, the results, and the conclusion. Below is part of my newly written summary. 

“In the visual design of a portal website, color is the first intuitive factor for users. It is relatively difficult for the designer of a portal website to choose a color system that represents a city's unique color among many colors. Therefore, this study extracted a decision-making model of the urban color system, which can help decision-makers and designers to choose among color systems, and then effectively design a portal website that conforms to local cultural attributes. The proposed method to solve the problem involved obtaining the optimal color matching by performing weight analysis on colors through 123 sample color semantics, factor analysis, and a fuzzy analytic hierarchy process. Semantic analysis was used to classify colors into four categories of fashion, technology, calm, and dazzling. The fashion color matching scheme scored relatively high. Web page color matching schemes with a white background were popular, among which white and green color matching schemes scored relatively high. At the same time, there are differences in color preferences between genders and cultures. This study is significant because it proposes a color decision model for portal websites, which provides a reference value that can also be applied to the selection of color schemes for other types of web pages in the future.”

• In the related work, it is recommended to refer the contribution made by the researchers and the novelty of the research. However, the author does not mention that.

Thank you very much for pointing out the need to highlight the contributions made by the researchers. I amended the article and emphasized the researcher's contribution in the abstract, methods, and conclusion of the article. In this context of digital multimedia, users' browsing habits are uncertain and random. At the same time, as a gateway to a city, a portal also has its own urban attributes, which contain physical geography, humanities, and development vision. Our contribution is to analyze the semantics of color to identify the four categories of fashion, technology, calmness, and dazzling. We used 123 samples, through fuzzy layer-by-layer AHP, and particle swarm optimization. The random and uncertain user habits of these users were discussed, and the color decision model was constructed that considered gender and different cultural backgrounds. This color decision model is actually an innovation in method, so in this case, our method and discussion results are our contribution.

• I recommend that the authors add some more current articles to improve the paper's overall quality. The preparation of a comparative analysis of the current publications on this subject should also be included.

Thank you very much for your suggestion. I re-quoted the latest article in the article and added a comparative analysis.

• Avoid presenting with lengthy paragraph.

Thank you very much for pointing out these problems with the article. I have re-adjusted and polished the article. We hired a professional editing agency to revise the errors in the manuscript and statement to increase the readability of the article.

• Paper needs to polish and provide a detailed explication of theoretical/systematic aspects behind this paper.

Thank you very much for pointing out this issue. We have retouched and polished the theoretical part of the article.

• Add more discussion on Color Behavior for that include quality research paper in the revised version.

Thank you for pointing out these issues. In the Discussion section I have included a comparative discussion of experimenting with color behavior. 

“For the evaluation of different color matching methods, comparison analysis of this color selection model and other models was used. Previous research used Colormind, a website that uses neural networks for color assignment. Various other methods have been used in research in the field of automatic color generation [1]. You Weitao [2] used a clustering method to select advertising colors, and obtained color preferences by collecting a certain number of advertising pictures. A limitation of this method is that only three types of advertisements can be determined when the selection of advertisements is too broad. Gu and Lou [3] proposed a web page automatic coloring model, analyzed the collected web page samples, and used a regression model to optimize the web page color blocks to obtain the optimal color. A limitation of this method is that because there are many types of websites, the scope is too large. To verify the effectiveness of the method, three different methods were tested. They are our own method, a data-driven method [3], and Colormind.”

“Additionally, we ran FAHP tests to analyze the effectiveness of different methods for producing final web pages. The same web page structure was prepared to extract the colors obtained by different methods for scoring. Five graphic design teachers and six laypeople were asked to rate the match between images and keywords using a 5-point Likert scale. The analysis mainly tested and scored the three methods for professionals and non-professionals. The scores obtained by the colors produced by our method are relatively high, and the same results can be seen from both the professional and the non-professional groups.”

1. Zhang R, Zhu JY, Isola P, Geng XY, Lin AS, Yu TH, et al. Real-time user-guided image colorization with learned deep priors. ACM Trans Graph. 2017;36: 1–11. doi: 10.1145/3072959.3073703. 

2. You WT, Sun LY, Yang ZY, Yang CY. Automatic advertising image color design incorporating a visual color analyzer. Comput Languages. 2019;55. doi: 10.1016/j.cola.2019.100910. 

3. Gu Z, Lou J. Data driven webpage color design. Comput Aid Des. 2016;77: 46-59. doi: 10.1016/j.cad.2016.03.001.

• Some more clarification regarding the motivation and challenges of the research. 

Thank you very much for pointing out this issue. We added further explanations on the motivation at the beginning of the introduction and made the following adjustments.

“On the Internet, a city portal is a window for delivering city images to online users irrespective of national geographic boundaries. Not only does it need to conform to local and native people's impressions of the city, it also needs to accurately convey the image of the city to users visiting the site for the first time. Therefore, the design of a city portal website should first highlight the city's unique culture, history, region, and other visual elements in the interface design, and at the same time, it should also be suitable for the user preferences of visitors from different countries.

In interface design, an effective interface layout can help users quickly obtain information. Color, as the visual identification element that accounts for the largest proportion of the interface area, and which is the first to catch the user's line of sight, will leave a deep impression on the user in the process of browsing the page. In addition, the Internet interface is different from the static screen, and the content of the Internet interface is updated from time to time. Therefore, we can only put constantly updated content and specifications in the corresponding position of the interface, so that the information will not be cluttered. Therefore, in the interface layout, color matching research about relatively stable colors and background colors occupying a large area in the interface can more accurately understand the color preferences of Internet users.”

• Notations and acronyms used in this paper should be summarized in a table to organize this paper in a better way.

Thank you very much for your comment. I have compiled and tabulated the abbreviations presented in this article. The abbreviations of some special vocabulary involved in this article are organized and summarized here to facilitate understanding of the full text.

Table 2 Fuzzy AHP score

Shorthand Full name

FAHP Fuzzy Analytic Hierarchy Process

PSO Particle Swarm Optimization

KMO Kaiser Meyer Olkin

SPSS Statistical Product Service Solutions

CR Consistency Ratio

• Improve the quality of figures and explain those properly.

Thank you for your comments on the figures. We redrew the figures and aimed to use a more intuitive design.

• Finally, a final proof-reading is highly suggested, in order to correct some typos.

We apologize for the poor language of our manuscript. We worked on the manuscript for a long time and the repeated addition and removal of sentences and sections obviously led to poor readability. We have now worked on both language and readability, and have also involved native English speakers for language corrections. We really hope that the flow and language level have been substantially improved.

---

## [Decision Letter · Decision Letter 1]

22 Feb 2023

The visual design of urban multimedia portals

PONE-D-22-14953R1

Dear Dr. Zhang,

We’re pleased to inform you that your manuscript has been judged scientifically suitable for publication and will be formally accepted for publication once it meets all outstanding technical requirements.

Kind regards,

Dhananjay Singh, Ph.D.

Academic Editor

PLOS ONE

Additional Editor Comments (optional):

Your work has been thoroughly reviewed, and we appreciate your diligent efforts to address the reviewers' comments and improve the quality of your manuscript. However, author need to proof-read their final version manuscript.

Reviewers' comments:

Reviewer's Responses to Questions

**Comments to the Author**

1. If the authors have adequately addressed your comments raised in a previous round of review and you feel that this manuscript is now acceptable for publication, you may indicate that here to bypass the “Comments to the Author” section, enter your conflict of interest statement in the “Confidential to Editor” section, and submit your "Accept" recommendation.

Reviewer #1: All comments have been addressed

Reviewer #2: All comments have been addressed

2. Is the manuscript technically sound, and do the data support the conclusions?

Reviewer #1: Yes

Reviewer #2: Yes

3. Has the statistical analysis been performed appropriately and rigorously? 

Reviewer #1: Yes

Reviewer #2: Yes

4. Have the authors made all data underlying the findings in their manuscript fully available?

Reviewer #1: Yes

Reviewer #2: Yes

5. Is the manuscript presented in an intelligible fashion and written in standard English?

Reviewer #1: (No Response)

Reviewer #2: Yes

6. Review Comments to the Author

Reviewer #1: The paper is well revised. The author has answered all the query and also incorporated the suggestions.

Reviewer #2: All the suggested changes are incorporated well. This paper can be considered now for publication.

7. PLOS authors have the option to publish the peer review history of their article (what does this mean?). If published, this will include your full peer review and any attached files.

Reviewer #1: No

Reviewer #2: No

---

## [Editor Report · Acceptance letter]

1 Mar 2023

PONE-D-22-14953R1 

The visual design of urban multimedia portals 

Dear Dr. Zhang:

I'm pleased to inform you that your manuscript has been deemed suitable for publication in PLOS ONE. Congratulations! Your manuscript is now with our production department. 

Kind regards, 

on behalf of

Dr. Dhananjay Singh 

Academic Editor

PLOS ONE